# Measuring the readiness to screen and manage intimate partner violence: Cross-cultural adaptation and psychometric evaluation of the PREMIS tool for perinatal care providers

**Candy Guiguet-Auclair**[1]*, **Anne Debost-Legrand**[1,2], **Didier Lémery**[1], **Chloé Barasinski**[1], **Blandine Mulin**[3], **Françoise Vendittelli**[1,2]

**1** Clermont Auvergne Université, CNRS, CHU Clermont-Ferrand, Institut Pascal, Clermont-Ferrand, France, **2** Auvergne Perinatal Health Network, CHU Clermont-Ferrand, Clermont-Ferrand, France, **3** Fédération Française des Réseaux de Santé Périnatale (FFRSP), Toulouse, France

\* cauclair@chu-clermontferrand.fr

**Data Availability Statement:** The data underlying this study have been uploaded to: Guiguet-Auclair,

## Abstract

### Introduction

Pregnancy and perinatal periods are significant risk factors of intimate partner violence (IPV), a major public health problem that could begin or intensify during these periods. Perinatal care providers have a major role in the identification and the management of IPV. This study aimed to cross-culturally adapt into French the Physician Readiness to Manage Intimate Partner Violence Survey (PREMIS) tool, a reliable instrument to assess the knowledge, attitudes and preparedness to address IPV, and to evaluate its psychometric properties.

### Methods

The PREMIS was cross-culturally adapted by conducting forward and backward translations, following international guidelines. An online cross-sectional study was conducted to assess the psychometric properties of the PREMIS-French in perinatal care providers: data completeness, factor analysis, score distribution, floor and ceiling effects, internal consistency, item-total correlations, inter-subscale correlations and test-retest reliability.

### Results

The PREMIS was successfully translated and cross-culturally adapted to the context of metropolitan France. The results obtained from 360 perinatal care providers showed good acceptability. Exploratory factor analysis of the "Opinions" items resulted in a six-factor solution with six of the eight subscales of the original structure identified. Good internal consistency (Cronbach's alpha ranging from 0.54 to 0.97) and good test-retest reliability (intraclass correlation coefficients ranging from 0.46 to 0.92) for the "Background" and "Opinions" subscales were found.

Candy; Debost-Legrand, Anne; Lémery, Didier; Barasinski, Chloé; Mulin, Blandine; Vendittelli, Françoise (2021), "Measuring the readiness to screen and manage intimate partner violence: cross-cultural adaptation and psychometric evaluation of the PREMIS tool for perinatal care providers", Mendeley Data, V2, doi: 10.17632/7x989dtpk7.2.

**Funding:** The authors received no specific funding for this work.

**Competing interests:** The authors have declared that no competing interests exist.

## Discussion

This study provides evidence of the good psychometric properties of the PREMIS-French. This valid instrument will help to understand perinatal care providers' barriers to IPV screening and management and will help to focus on specific lacks of knowledge for developing IPV education programs.

## Introduction

Intimate partner violence (IPV) is well recognized as a major public health problem [1]. IPV prevalence in Europe is estimated to exceed 20% in women aged 15 years and older and is higher in women of reproductive age [2]. Pregnancy and perinatal periods are significant risk factors of IPV that could begin or intensify during these periods. In 2013, a meta-analysis reported a mean IPV prevalence rate among pregnant women of 19.8% for 23 countries; the mean reported prevalence of abuse was 28.4% for emotional abuse, 13.8% for physical abuse and 8.0% for sexual abuse [3]. Acts of violence during pregnancy affects directly and indirectly the mortality and the morbidity of the child and the mother. IPV is associated to unwanted pregnancies, termination of pregnancy [4] and adverse maternal and neonatal outcomes [5–7]. Moreover, the prevalence of maternal death by homicide was estimated between 13 and 24%, considering all causes of deaths during pregnancy [8].

Given these serious health consequences, healthcare professionals specializing in the perinatal period (perinatal care providers), have a major role in the identification and the management of abused women. The French National Health Authority recommended in 2019 that perinatal care providers screen all women for IPV during the first prenatal visit and continue at least once per trimester and during postpartum visits [9], in accordance with international guidelines [10]. Pregnancy is recognized as a privileged moment to detect past or current violence suffered by a woman. Indeed, the frequency of prenatal consultations and the bond of trust that is established between the care provider and the women offer the opportunity to disclose IPV.

However, several studies shown that IPV screening was not always done by healthcare professionals, often in reason of insufficient preparedness [11–13]. In 2014, the French inter-ministerial mission for the protection of women victims of violence and the fight against human trafficking conducted a national study to evaluate the midwives' IPV knowledge and their ability to screen IPV with a non-validated questionnaire [14]. They reported insufficient education on IPV and limited knowledge. Almost one third had never assessed their patients' exposure to IPV and in case of IPV, 81% felt insufficiently prepared. Nonetheless, 90% considered that midwives had a role to play in the disclosure and management of IPV. The reasons given for not screening were the lack of training, the absence of clear referral procedures and the ignorance of appropriate response or local specialist IPV services. Similar results were found among Italian midwives [15]. An American study identified in 2015 that 31.2% of obstetricians-gynecologists screen all pregnant patients at specific times of their pregnancy and 36.8% report insufficient training to assist their patients in addressing IPV [16].

Thus, the creation of specific IPV education and training programs for perinatal care providers, teaching how to recognize and respond to IPV is primordial [17,18], as well as assessing the effectiveness of these educational interventions with reliable outcomes. Among them, the PREMIS (Physician Readiness to Manage Intimate Partner Violence Survey), developed by Short et al. in 2006, is a valid and reliable self-reported instrument designed to measure the knowledge, attitudes and preparedness of physician to manage IPV [19]. They demonstrated

evidence for its construct validity, internal consistency (with Cronbach's alpha higher than 0.65 for reliable subscales), predictive validity, external validity (correlations between self-reported practices and observed clinical behaviors) and temporal stability over 12 months in the absence of IPV education or intervention. The developers shown that the PREMIS could discriminate between trained and untrained physicians [20].

This instrument has been adapted and widely used in multiple populations, cultures and languages [16,21–33]. It was also used to measure the effectiveness of IPV training programs proposed to physicians [20], practicing general practitioners and residents of general practice [34], nurses [35], and healthcare providers working in fracture clinics [36].

To our knowledge, no study in France assessed with a valid and reliable tool neither IPV knowledge of perinatal care providers nor their ability to identify and address IPV victims. Thus, we performed a study to cross-culturally adapt and evaluate the psychometric properties of a French version of the PREMIS tool.

## Methods

### Study design and participants

This cross-sectional online study was conducted from May 2017 to December 2019 in two phases: first, the original PREMIS was cross-culturally adapted from English to French; then psychometric properties of the French version of PREMIS were assessed.

The study was approved by the French regional ethics committee "*Comité d'Ethique des Centres d'Investigation Clinique de l'inter-région Rhône-Alpes-Auvergne, CE-CIC Grenoble*" (n° IRB 0005921, 03 October 2016). The aims and procedures were explained to participants, who gave their written informed consents.

An online link was created and data were collected using REDCap electronic data capture tools hosted at Clermont-Ferrand University Hospital. The French Federation of Perinatal Heath Network ('*Fédération Française des Réseaux de Santé en Périnatalité*') invited by email its care providers' memberships (except healthcare students) to take part in the study and spread the internet link to complete the questionnaire. A reminder was done one time during the period of the study.

To assess test-retest reliability of the PREMIS, the internet link to complete the questionnaire a first time was sent by email to a subsample of memberships of two regional perinatal networks who were identified by their email address. They completed a second time the same questionnaire online between fifteen days and one month after the first assessment. Respondents who reported having a training course about IPV between test and retest were excluded for this reliability analysis.

The sample size of the study was determined according to quality criteria established by COSMIN [37] and Terwee et al. [38] that recommend a minimum number of 100 subjects to ensure satisfactory factor analysis and internal consistency evaluation and a sample size of at least 50 subjects in order to guarantee an acceptable assessment for reliability.

### The PREMIS tool

The PREMIS self-administrated questionnaire contains 67 items comprising five sections: "Respondent profile", "Background", "Actual IPV Knowledge", "Opinions" and "Practice issues".

From the "Background" section, two subscales scores are obtained by calculating the mean of the individual scores of the items listed in the subscale. The 'Perceived preparation' subscale includes 12 items asking respondents how prepared they feel to assist IPV victims and rated on a 7-point Likert scale from 1 (not prepared) to 7 (quite well prepared). The 'Perceived

knowledge' subscale includes 16 items asking respondents how much respondents feel they know about IPV and rated on a 7-point Likert scale from 1 (nothing) to 7 (very much).

In the "Actual IPV Knowledge" section, a combination of 7 multiple choice items and 11 true/false items (on IPV risk, manifestations, victim's behavior and physician's professional practice) form the 'Actual knowledge' subscale, scored by the number of correct items.

The "Opinions" section includes 36 items, rated on a 7-point Likert scale from 1 (strongly disagree) to 7 (strongly agree) and evaluating attitudes, beliefs and physician's role in IPV management, on which 31 items forms six reliable subscales in the original version of PREMIS: 'Preparation', 'Legal requirements', 'Workplace issues', 'Self-efficacy', 'Alcohol/drugs' and 'Victim understanding'. Two other subscales, 'Constraints' and 'Victim autonomy', needed future testing for validity and reliability.

Scores are obtained by calculating the mean of the individual scores of the items listed in the subscale (provided that answers are given to at least half of the items), with reversed scores for negatively worded opinion items.

The 13 items of the "Practice issues" section evaluated self-reported behaviors and personal IPV experience of the respondent, with various responding options.

## Translation and cross-cultural adaptation of the French version of PREMIS

The PREMIS questionnaire was translated from English into French and cross-culturally adapted to be relevant to the French context, following international guidelines for the adaptation of self-administered instruments [39].

Forward translations were independently made by two bilingual translators fluent in English, with French as mother tongue, and naïves to the outcome measure. A multidisciplinary expert committee (composed of an obstetrician-gynecologist practicing forensic assessments, an obstetrician-gynecologist supervising a perinatal network, a referent midwife for IPV, and methodologists) reviewed the two translations and edited a first consensus French version. Cultural adaptations and linguistic equivalence with the original English version of PREMIS were discussed.

Then a native English translator fluent in French, blinded to the original English version then made a backward translation. The expert committee compared source and target versions, and resolved discrepancies. Item translation, semantic, idiomatic, cultural, experiential, and conceptual equivalents were discussed. The consensus French version was pre-tested on a sample of ten health care providers (general practitioners, obstetrician-gynecologists, and midwives) in order to evaluate the comprehensibility of instructions and items. Their responses and comments were reviewed.

The evaluation of psychometric properties (given in details below) was then conducted.

## Statistical analysis

Statistical analysis were performed with SAS software (version 9.4, SAS Institute, Cary, NC, 2002–2012) and conducted at a two-sided alpha = 0.05 significance level.

Sociodemographic and professional characteristics of participants were described.

The psychometric properties evaluation of the PREMIS-French consisted in data completeness, factor analysis, descriptive statistics and score distributions, internal consistency, item-total correlations, inter-subscale correlations and reliability.

**Data completeness.** The respondent acceptability was assessed by looking at the frequency of missing values. Data quality was considered satisfactory if less than 15% of the item data were missing.

**Factor analysis.** Factor analysis with an oblique promax rotation, allowing the factors to correlate, were performed to study the multidimensionality and distribution of the items of the "Opinions" section in subscales [40]. As attitudes and clinical practice in IPV management could vary from culture to culture, there was no guarantee that the French version reproduced the subscales of the original PREMIS questionnaire. Hence, we chose an exploratory analysis of the structure of the items [39,41–43]. The Kaiser-Meyer-Olkin (KMO) statistic and Bartlett's test of sphericity were used to check the appropriateness of running the factor analysis. KMO values higher than 0.50 are acceptable [44]. Bartlett's test requires to yield significant result ($p < 0.05$). Eigenvalues higher than 1 (Kaiser criterion) and Cattell's scree plot [45] were used for factor retention. The solution giving the most adequate factor structure (item loadings greater than 0.32, no or few item cross loadings, i.e. no or few items with loadings at 0.32 or higher on two or more factors) was retained [40].

**Descriptive statistics and score distributions.** The PREMIS-French subscales scores' distribution were described by mean, standard deviation, median and range. The variability of the PREMIS-French scores was investigated for each subscale with the floor and ceiling effects. These effects were considered to be present if more than 15% of the subjects obtained the lowest or highest possible score [46].

**Internal consistency.** Cronbach's α coefficient was used to evaluate the internal consistency of each subscale of the "Background" and "Opinions" sections [47]. The minimum required for the coefficient was 0.70, according to the standard used for group comparisons [48]. Internal consistency was not evaluated for the 'Actual knowledge' subscale as it is a criterion-referenced subscale [19,49].

**Item-total correlations.** Item-total consistency was used to evaluate the extent of the linear relationship between an item and its subscale for the "Background" and "Opinions" sections, corrected for overlap (the item which is to be correlated with the scale was omitted from the subscale total) [43]. A minimum correlation coefficient of 0.40 was considered indicative of good item-total consistency [50].

**Inter-subscale correlations.** Spearman's coefficients were used to evaluate inter-subscale correlations of the "Opinions" section subscales. Correlations were considered very small for coefficients lower than 0.30, small for coefficients between 0.30 and 0.50, moderate from 0.50 to 0.70 and strong if higher than 0.70 [51].

**Convergent validity.** Spearman's coefficients were used to evaluate correlations between (1) the number of hours of previous IPV training, 'Perceived knowledge', 'Perceived preparation', and 'Actual knowledge' subscales, (2) the subscales from the "Opinions" section and the number of hours of previous IPV training, 'Perceived knowledge', 'Perceived preparation', 'Actual knowledge' subscales. Positive correlations were expected.

**Reliability.** Stability over time was assessed by the test-retest method. Reliability of the subscales from the "Background", "Actual IPV knowledge" and "Opinions" sections was estimated by intraclass correlation coefficient (ICC), based on the two-way mixed effect model. The following categories were selected to interpret the agreement levels: 0–0.2 small, 0.21–0.40 fair, 0.41–0.60 moderate, 0.61–0.80 substantial and 0.81–1 almost perfect [52].

## Results

### Translation and cross-cultural adaptation

Cultural adaptations were made during the forward translations. In "Respondent profile" section, an item was added: '*Including you, how many midwives practice at your work site?*'. In "Actual IPV Knowledge" section, '*Child Protective Services*' was adapted to the French context using '*Cellule de Recueil des Informations Préoccupantes (CRIP)*'. In "Opinions" section, item

17 ('*I comply with the Joint Commission standards that require assessment for IPV*') was adapted to the French context, as the Joint Commission did not exist in France. In "Practice issues" section, one item was not culturally relevant in the French context and was removed (item 8: '*Do you practice in a state where it is legally mandated to report IPV cases involving competent (non-vulnerable) adults*?'). Moreover, local DV/IPV hotline and Child Protective Services were omitted from the list of referral resources in item 4, as these support services are not available in France.

In pre-testing the French version, none of the ten health care providers reported any understanding difficulty or completion's problem. Consequently, the expert committee adopted this version as the pre-final cross-cultural adaptation (S1 File). We named this version the PREMIS-French.

## Psychometric evaluation of the PREMIS-French

**Participants.** The online survey was answered by 360 perinatal care providers. The characteristics of these respondents are described in Table 1 and S1 Table. They were predominantly women with a mean age of 43.3 years (SD 10.7) and practicing for 17.9 years (SD 10.8) mainly in gynecology and obstetrics. Less than 4% had no actual clinical activity. About one

**Table 1. Sociodemographic and professional characteristics of participants.**

| Participants' characteristics | N = 360 |
|---|---|
| Age (years), *n (%)* | |
| 20–29 | 32 (8.9) |
| 30–39 | 112 (31.1) |
| 40–49 | 97 (26.9) |
| 50–59 | 93 (25.8) |
| 60–69 | 26 (7.2) |
| Women, *n (%)* | 337 (93.9) |
| Field of practice, *n (%)* | |
| Internal medicine | 1 (0.3) |
| General practitioner | 25 (6.9) |
| Pediatrics | 44 (12.2) |
| Psychiatry | 4 (1.1) |
| Surgery | 1 (0.3) |
| Gynecology and obstetrics | 244 (67.8) |
| Mother and child protection services | 29 (8.1) |
| Other | 32 (8.9) |
| Number of years practicing, *mean ± SD* | 17.9 ± 10.8 |
| Total number of hours of previous IPV training, *n (%)* | |
| None | 85 (26.6) |
| < 10h | 117 (36.7) |
| 10h-19h | 56 (17.6) |
| ≥ 20h | 61 (19.1) |
| IPV experience in the last 6 months *, *n (%)* | |
| Yes | 205 (63.7) |
| No | 117 (36.3) |

* At least one new diagnose of IPV (picked up an acute case, uncovered ongoing abuse, or had a patient disclose a past history) made in the last 6 months.

quarter of the care providers never had training about IPV issues. The mean amount of previous IPV training was 15.0 hours (SD 19.3) and the more frequent type of training was to attend a lecture or talk. In the last 6 months, 63.7% made at least one new diagnose of IPV (picked up an acute case, uncovered ongoing abuse, or had a patient disclose a past history).

**Data completeness.** In the "Respondent profile" section, the percentage of missing values per item varied between 0 and 16.1% (for the item on the number of practitioners that have participated in an IPV training course in the past 6 months at the work site). In the "Background" section, the percentage of missing values was 11.4% for the item on the amount of previous IPV training and ranged from 0 to 1.7% for items of the 'Perceived preparation' subscale and from 0.6 to 2.2% for items of the 'Perceived knowledge' subscale. The percentage of missing values ranged from 0.3 to 2.8% for items of the "Actual IPV knowledge" section and from 1.9 to 7.8% for items of the "Opinion" section. In the "Practice issues" section, with items on the screening and management of IPV victims in the last 6 months, the percentage of missing values per item were higher and ranged between 4.2 and 16.4%.

**Factor analysis.** Factor analysis with an oblique promax rotation were performed to study the distribution of the 36 items of the "Opinions" section in subscales. The significance value of Bartlett's test of sphericity was <0.0001 ($\chi^2$ = 3606.5, $df$ = 630) and KMO measures of sampling adequacy was 0.846, indicating that the data were suitable for factor analysis. An initial Maximum Likelihood Factor (MLF) analysis identified a 11-factor solution that was statistically sound ($\chi^2$ = 323.7, $df$ = 289, $p$ = 0.0783). However, four factors contained only two items, ten items cross-loaded on two or more factors, indicating a complex solution that lacked a good theoretical basis. Thus, new factor analysis were performed after discarding items with loadings lower than 0.32, with loadings greater than 0.32 on two or more factors or items that significantly lowered internal consistency. The final factor analysis identified six factors with eigenvalues greater than one and accounting for 59.1% of the total variance (Table 2). All items loaded higher than 0.40 on its subscale. The first factor comprised five items and was named 'Preparation'. It had four items in common with the original 'Preparation' subscale (items 6, 10a, 10b and 10c) and one item from the original 'Self-efficacy' subscale (item 9). The second factor named 'Workplace issues' comprised six items, four items were in common with the original subscale (items 3, 4, 19, 25), item 2 was from the original 'Self-efficacy' subscale and item 18 was from the original 'Victim understanding' subscale. The third factor comprised the four items of the original 'Legal requirements' subscale (items 12a, 12b, 12c, 17) and was labeled identically. Item 17 loaded higher than 0.32 on two factors: 0.55 on its factor and 0.38 on factor 2. The fourth factor comprised four items (items 5, 14, 26, 32) and was labeled 'Self-efficacy'. It had three items in common with the original subscale (items 5, 14, 32) and one item from the original 'Workplace issues' subscale (item 26). The fifth factor comprised the three items of the original 'Alcohol/drugs' subscale (items 7, 21, 31) and was named identically. The sixth factor comprised three items of the original 'Victim understanding' subscales (items 11, 15, 16) and was labeled identically. Item 15 loaded higher than 0.32 on two factors: 0.71 on its factor and 0.33 on factor 4.

Two original "Opinions" subscales were not found in the current model. The items of 'Victim autonomy' (items 8, 22, 30) and 'Constraints' (items 13 and 23) subscales were discarded during the selection process of the most adequate factor structure.

**Descriptive statistics, score distribution, floor and ceiling effects.** The descriptive statistics and score distributions of the PREMIS-French subscales are presented in Table 3. The percentage of missing values were less than 7%, except for the 'Actual knowledge' subscale with 25% of missing values. Neither floor nor ceiling effects were found for the "Background", "Actual IPV knowledge" and "Opinions" subscales.

**Table 2. Factor loadings from the factor analysis of the PREMIS-French "Opinions" items.**

| | Factor 1 | Factor 2 | Factor 3 | Factor 4 | Factor 5 | Factor 6 |
|---|---|---|---|---|---|---|
| **Variance explained (%)** | **27.5** | **8.1** | **6.7** | **6.4** | **5.5** | **4.9** |
| **'Preparation' subscale** | | | | | | |
| 6. Do not have sufficient training to assist individuals in addressing IPV situations | **0.54** | 0.09 | 0.07 | 0.07 | -0.14 | 0.00 |
| 9. Feel comfortable discussing IPV with patients | **0.50** | 0.29 | 0.06 | 0.21 | -0.02 | 0.04 |
| 10a. Do not have the necessary skills to discuss abuse with a female IPV victim | **0.94** | 0.03 | -0.05 | -0.02 | 0.02 | 0.00 |
| 10b. Do not have the necessary skills to discuss abuse with a male IPV victim | **0.90** | -0.15 | -0.01 | -0.01 | 0.04 | -0.01 |
| 10c. Do not have the necessary skills to discuss abuse with an IPV victim from a different cultural/ethnic background | **0.98** | -0.01 | -0.09 | -0.06 | 0.01 | 0.01 |
| **'Workplace issues' subscale** | | | | | | |
| 2. Ask all new patients about abuse in their relationships | 0.01 | **0.84** | -0.14 | 0.00 | 0.03 | 0.10 |
| 3. Workplace encourages to respond to IPV | 0.09 | **0.60** | 0.22 | -0.12 | -0.09 | -0.01 |
| 4. Can make appropriate referrals to services within the community for IPV victims | 0.25 | **0.41** | 0.21 | -0.12 | 0.01 | 0.00 |
| 18. Health care providers have a responsibility to ask all patients about IPV | -0.06 | **0.87** | -0.18 | -0.10 | 0.11 | 0.06 |
| 19. Practice setting allows adequate time to respond to victims of IPV | 0.02 | **0.43** | 0.17 | 0.22 | -0.05 | -0.12 |
| 25.Adequate private space to provide care for victims of IPV | -0.05 | **0.45** | 0.16 | 0.18 | -0.04 | -0.13 |
| **'Legal requirements' subscale** | | | | | | |
| 12a. Aware of legal requirements regarding reporting of IPV suspected cases | 0.09 | 0.14 | **0.68** | 0.07 | -0.07 | 0.11 |
| 12b. Aware of legal requirements regarding reporting of child abuse suspected cases | 0.05 | 0.07 | **0.81** | -0.11 | 0.04 | 0.01 |
| 12c. Aware of legal requirements regarding reporting of elder abuse suspected cases | -0.04 | -0.26 | **0.85** | 0.07 | 0.06 | 0.00 |
| 17. Comply with the law standards that require reporting for IPV | -0.19 | **0.38** | **0.55** | -0.03 | 0.02 | 0.02 |
| **'Self-efficacy' subscale** | | | | | | |
| 5. Capable of identifying IPV without asking patient about it | 0.03 | -0.14 | 0.06 | **0.77** | 0.08 | -0.01 |
| 14. Able to gather the necessary information to identify IPV as the underlying cause of patient illnesses | 0.25 | 0.18 | 0.08 | **0.48** | 0.02 | -0.03 |
| 26. Able to gather the necessary information to identify IPV as the underlying cause of patient injuries | 0.18 | 0.31 | 0.09 | **0.44** | 0.03 | -0.08 |
| 32. Can recognize victims of IPV by the way they behave | -0.08 | -0.02 | -0.11 | **0.83** | 0.09 | 0.06 |
| **'Alcohol/drugs' subscale** | | | | | | |
| 7. Patients who abuse alcohol or other drugs are likely to have a history of IPV | 0.01 | 0.15 | 0.02 | 0.09 | **0.69** | 0.04 |
| 21. Alcohol abuse is a leading cause of IPV | 0.08 | -0.16 | 0.12 | 0.00 | **0.76** | 0.03 |
| 31. Use of alcohol or other drugs is related to IPV victimization | -0.15 | 0.08 | -0.08 | 0.14 | **0.69** | -0.04 |
| **'Victim understanding' subscale** | | | | | | |
| 11. If victims of abuse remain in the relationship after repeated episodes of violence, they must accept responsibility for that violence | -0.02 | 0.07 | 0.03 | -0.08 | 0.03 | **0.69** |
| 15. If a patient refuses to discuss the abuse, health care providers can only treat the patient's injuries | -0.06 | -0.07 | -0.04 | **0.33** | -0.25 | **0.71** |
| 16. Victims of abuse could leave the relationship if they wanted to | 0.07 | 0.01 | 0.08 | -0.13 | 0.18 | **0.73** |

Loadings equal or higher than 0.32 are presented in bold.

**Internal consistency and item-total correlations.** The "Background" subscales showed good internal consistency, with Cronbach's α equal to 0.95 and 0.97 for 'Perceived preparation' and 'Perceived knowledge' subscales respectively. All corrected item-total correlations were higher than the required 0.40, ranging from 0.54 to 0.87 for 'Perceived preparation' subscale and from 0.50 to 0.90 for 'Perceived knowledge' subscale.

For "Opinions" subscales, Cronbach's α ranged from 0.54 to 0.88 (Table 4), showing good internal consistency for four subscales, and moderate internal consistency for the 'Alcohol/drugs' and 'Victim understanding' subscales, which did not obtain the minimum required coefficient of 0.70. All corrected item-total correlations were higher than the required 0.40, ranging from 0.40 to 0.85, except for item 21 from 'Alcohol/drugs' subscale with a value of

**Table 3. Descriptive statistics and score distributions of the PREMIS-French subscales.**

| PREMIS-French subscales | Missing values (%) | Mean ± SD | Range | Median | Floor effect (%) | Ceiling effect (%) |
|---|---|---|---|---|---|---|
| Background | | | | | | |
| Perceived preparation | 0 | 3.47 ± 1.28 | 1.00–7.00 | 3.42 | 1.39 | 0.56 |
| Perceived knowledge | 0.56 | 3.59 ± 1.27 | 1.25–7.00 | 3.56 | 0 | 0.56 |
| Actual IPV knowledge | | | | | | |
| Actual knowledge | 25 | 25.44 ± 4.77 | 0–38.00 | 26.00 | 0 | 0 |
| Opinions | | | | | | |
| Preparation | 3.33 | 4.39 ± 1.46 | 1.00–7.00 | 4.55 | 1.72 | 2.01 |
| Workplace issues | 1.67 | 4.58 ±1.19 | 1.40–7.00 | 4.50 | 0 | 1.13 |
| Legal requirements | 3.89 | 3.87 ±1.30 | 1.00–7.00 | 3.75 | 1.16 | 0.87 |
| Self-efficacy | 3.89 | 3.66 ±1.10 | 1.00–7.00 | 3.75 | 0.29 | 0.29 |
| Alcohol/drugs | 6.11 | 4.30 ±1.00 | 1.00–7.00 | 4.33 | 0.30 | 0.89 |
| Victim understanding | 4.72 | 5.37 ±1.04 | 2.67–7.00 | 5.33 | 0 | 9.62 |

0.38, and for items 11 and 15 from 'Victim understanding' subscale with values of 0.32 and 0.34 (S2 Table).

In addition, each item of the "Opinions" section correlated better with its parent subscale (corrected for overlap) than with the other subscales, except for two items of the 'Self-efficacy' subscale (S2 Table). For item 14 (*Able to gather the necessary information to identify IPV as the underlying cause of patient illnesses*), the item-total correlation with its subscale was 0.57 and the correlation with the 'Preparation' subscale was also 0.57. For item 26 (*Able to gather the necessary information to identify IPV as the underlying cause of patient injuries*), the item-total correlation with its subscale was 0.54 and the correlation with the 'Workplace issues' subscale was 0.57.

**Inter-subscale correlations.** Correlations between PREMIS-French "Opinions" subscales ranged from -0.003 to 0.54 (Table 4). The 'Preparation' subscale had a significant moderate correlation with the 'Workplace issues' subscale (r = 0.54). The 'Legal requirements' subscale had significant small correlations with 'Preparation' (r = 0.44), 'Workplace issues' (r = 0.45) and 'Self-efficacy' (r = 0.45) subscales. The 'Self-efficacy' subscale had significant moderate correlations with 'Preparation' (r = 0.53) and 'Workplace issues' (r = 0.50) subscales. The

**Table 4. Internal consistency (Cronbach's α) and inter-subscale correlations for the PREMIS-French "Opinions" subscales.**

| PREMIS-French "Opinions" subscales | Preparation | Workplace issues | Legal requirements | Self-efficacy | Alcohol/drugs | Victim understanding |
|---|---|---|---|---|---|---|
| Preparation | **0.88** | | | | | |
| Workplace issues | 0.54 *** | **0.77** | | | | |
| Legal requirements | 0.44 *** | 0.45 *** | **0.76** | | | |
| Self-efficacy | 0.53 *** | 0.50 *** | 0.45 *** | **0.72** | | |
| Alcohol/drugs | -0.003 | 0.14 * | 0.10 | 0.17 ** | **0.59** | |
| Victim understanding | 0.21 *** | 0.13 * | 0.09 | 0.09 | 0.02 | **0.54** |

Adapted from Short et al. [19].

Cronbach's α are reported on the diagonal and in bold text.

Spearman's correlation coefficients significantly different from zero:

* p<0.05,

** p<0.01 and

*** p<0.001.

**Table 5. Spearman's correlation coefficients between the amount of previous IPV training and the PREMIS-French "Background", "Actual IPV knowledge", "Opinions" subscales.**

| | | Background | | Actual IPV knowledge |
|---|---|---|---|---|
| | **Hours of previous IPV training** | **Perceived preparation** | **Perceived knowledge** | **Actual knowledge** |
| Background | | | | |
| Perceived preparation | 0.66 *** | 1 | | |
| Perceived knowledge | 0.67 *** | 0.91 *** | 1 | |
| Actual IPV knowledge | | | | |
| Actual knowledge | 0.28 *** | 0.22 *** | 0.27*** | 1 |
| Opinions | | | | |
| Preparation | 0.52 *** | 0.70 *** | 0.74 *** | 0.18 ** |
| Workplace issues | 0.52 *** | 0.52 *** | 0.54 *** | 0.18 ** |
| Legal requirements | 0.45 *** | 0.61 *** | 0.64 *** | 0.17 ** |
| Self-efficacy | 0.46 *** | 0.62 *** | 0.61 *** | 0.23 *** |
| Alcohol/drugs | 0.05 | -0.004 | 0.01 | 0.14 * |
| Victim understanding | 0.11 | 0.15 ** | 0.18 *** | 0.33 *** |

Adapted from Short et al. [19].

Correlations significantly different from zero:

* p<0.05,

** p<0.01 and

*** p<0.001.

'Alcohol/drugs' subscales had significant very small correlations with 'Workplace issues' (r = 0.14) and 'Self-efficacy' (r = 0.17) subscales, and no significant correlation with the others. The 'Victim understanding' subscale had only significant very small correlations with 'Preparation' (r = 0.21) and 'Workplace issues' (r = 0.13) subscales.

**Convergent validity.** Correlations between the amount of previous IPV training, the "Background" subscales, the 'Actual knowledge' subscale and the "Opinions" subscales are shown in Table 5.

The amount of previous IPV training was moderately correlated to 'Perceived preparation' and 'Perceived knowledge' subscales (r = 0.66 and r = 0.67, respectively) and very weakly to 'Actual knowledge' subscale (r = 0.28). The 'Perceived knowledge' and 'Perceived preparation' subscales were strongly correlated (r = 0.91). Correlations between the 'Actual knowledge' subscale and the "Background" subscales were significant but very small: r = 0.22 with 'Perceived preparation' and r = 0.27 with 'Perceived knowledge'.

Hours of IPV training showed moderate correlations with 'Preparation' (r = 0.52) and 'Workplace issues' (r = 0.52) subscales, almost moderate correlations with 'Legal requirements' (r = 0.45) and 'Self-efficacy' (r = 0.46) subscales, and no significant correlation with the two other "Opinions" subscales. All "Opinions" subscales except for the 'Alcohol/drugs' one were significantly correlated to the 'Perceived preparation' and 'Perceived knowledge' subscales. Correlations were strong with the 'Preparation' subscale (r = 0.70 and 0.74, respectively), moderate with 'Workplace issues', 'Legal requirements' and 'Self-efficacy' subscales (ranging from 0.52 to 0.62 and from 0.54 to 0.64, respectively) and very small for the 'Victim understanding' subscale (r = 0.15 and 0.18, respectively). Correlations between "Opinions" subscales and the 'Actual knowledge' subscale were significant but very small, ranging from 0.14 to 0.23, except for the 'Victim understanding' subscale with a higher correlation of 0.33.

**Reliability.** Of the fifty perinatal care providers randomly selected for the test-retest, 40 (80.0%) completed the test questionnaire. Among them, 24 (60.0%) completed the retest

**Table 6. Test-retest reliability for the PREMIS-French "Background", "Actual IPV knowledge" and "Opinions" subscales.**

| PREMIS-French subscales | ICC (95% CI) |
|---|---|
| Background | |
| Perceived preparation | 0.92 (0.83–0.97) |
| Perceived knowledge | 0.88 (0.74–0.95) |
| Actual IPV knowledge | |
| Actual knowledge | 0.69 (0.27–0.89) |
| Opinions | |
| Preparation | 0.65 (0.35–0.83) |
| Workplace issues | 0.78 (0.54–0.90) |
| Legal requirements | 0.77 (0.55–0.89) |
| Self-efficacy | 0.83 (0.62–0.93) |
| Alcohol/drugs | 0.46 (0.07–0.73) |
| Victim understanding | 0.74 (0.48–0.88) |

ICC (95% CI): Intraclass correlation coefficient (95% confidence interval).

questionnaire. No one reported having a training course about IPV between test and retest. The characteristics of the test-retest participants are described in S1 Table. They were predominantly women (83.3%), with a mean age of 44.2 years (SD 11.6). Twenty of them practiced in gynecology and obstetrics, two in mother and child protection services, one in pediatrics and one in psychology.

Test-retest reliability was almost perfect for the "Background" subscales, with ICCs equal to 0.92 for 'Perceived preparation' and 0.88 for 'Perceived knowledge' (Table 6). Agreement was substantial for 'Actual knowledge' subscale with an ICC of 0.69. Reliability was also substantial for "Opinions" subscales, with ICCs ranging from 0.65 to 0.83, except for 'Alcohol/drugs' subscale with a moderate ICC of 0.46.

## Discussion

The present study describes the cross-cultural adaptation and the evaluation of the psychometric properties of the French version of the PREMIS, named the PREMIS-French, in a national sample of perinatal care providers.

The PREMIS was successfully translated and cross-culturally adapted from English to French. The PREMIS-French had good acceptability with low percentages of missing values per item, except for some items from the "Practice issues" section, evaluating practices or workplace specificities that could be not relevant for some participants. It also had good response distribution, neither floor nor ceiling effect were found for the subscales, indicating that the instrument was adapted to the studied population.

The six-factor structure of the PREMIS-French "Opinions" section differs slightly from the eight-factor structure of the original version of the PREMIS [19]. Only the two original 'Victim autonomy' and 'Constraints' subscales were not found in our validation. However, in the original PREMIS, this subscales were not sufficiently reliable and only kept for future testing.

The 'Preparation' subscale groups together four items from the original 'Preparation' subscale and one item from the original 'Self-efficacy' subscale. This item (*Feel comfortable discussing IPV with patients*) was more related to the feeling of being prepared in the French context. The PREMIS-French 'Workplace issues' subscale comprised items from the original subscale, and two items related to the systematic abuse screening of patients from the original

'Self-efficacy' and 'Victim understanding' subscales. IPV screening in France depends on the institutions' policies where health care providers work and not all encourage IPV identification unfortunately. The 'Self-efficacy' subscale comprised one item from the original 'Workplace issues' subscale. Being able to gather the necessary information to identify IPV as the underlying cause of patient injuries is more related to self-efficacy than to work site. The 'Victim understanding' subscale comprised three items on the seven items of the original subscale. Lastly, 'Legal requirements' and 'Alcohol/drugs' subscales were identical in the French and the original versions [19].

The other PREMIS versions also presented factor structures that were somewhat different from the original version. Two subscales were identified in all other studies, 'Preparation' and 'Alcohol/drugs' subscales. The Greek physicians' version [23] identified a 2-item 'IPV screening' subscale. The 'Legal requirements' subscale was not found in this version where the items were excluded during the adaptation of the PREMIS [23], as the 'Victim autonomy' subscale. The Spanish version for physicians and nurses identified a 'Barriers' subscale [24]. The American pharmacists' version identified one single subscale for self-efficacy and workplace-efficacy [22]. The American health care students' version identified a 2-item 'IPV screening' subscale [21]. The 'Workplace issues' and the 'Constraints' subscales were not identified, as it was not appropriate for students [21]. The Australian paramedic and nursing students' version did not find the 'Workplace issues' and 'Constraints' subscales [25]. The 'Legal requirements' subscale was not found in this version as the items were excluded during the adaptation of the PREMIS [25].

Internal consistency of the "Background" subscales was high and comparable to those of the original subscales with Cronbach's $\alpha$ superior to 0.90 [19]. Same results were found for the physicians' Greek and Spanish versions [23,24], the American and Australian healthcare students' versions [21,25] and the American pharmacists' version [22]. For "Opinions" section, 'Preparation', 'Workplace issues', 'Legal requirements' and 'Self-efficacy' subscales demonstrated good internal consistency ($\alpha = 0.88$, $\alpha = 0.77$, $\alpha = 0.76$, and $\alpha = 0.72$), comparable to those reported in the original version ($\alpha = 0.85$, $\alpha = 0.82$, $\alpha = 0.79$, and $\alpha = 0.69$ respectively) [19]. 'Alcohol/drugs' and 'Victim understanding' subscales displayed moderate internal consistency ($\alpha = 0.59$ and $\alpha = 0.54$), lower than those reported in the original version ($\alpha = 0.70$ and $\alpha = 0.69$ respectively) [19]. This moderate internal consistency was also found in the Greek physicians' version ($\alpha < 0.50$ and $\alpha = 0.63$ respectively) [23], in the American healthcare students' version ($\alpha = 0.48$ and $\alpha = 0.46$ respectively) [21], and in the Australian paramedic and nursing students' version ($\alpha = 0.57$ for the 'Alcohol/drugs' subscale) [25]. The Spanish physicians and the American pharmacists' versions reported higher internal consistency for the 'Alcohol/drugs' subscale ($\alpha = 0.66$ and $\alpha = 0.80$ respectively) [22,24].

The very small to moderate correlations between the "Opinions" subscales imply that they measures related but relatively different constructs. 'Alcohol/drugs' subscale displayed the lowest correlations with the other subscales, results found in the original development of the PREMIS [19] and in the other studies [21–25].

Convergent validity was explored by assessing the inter-subscales correlations. The majority of the correlations replicated findings of previous studies. Only the Spanish validation did not explore these correlations [24]. 'Perceived preparation' and 'Perceived knowledge' subscales were strongly correlated ($r = 0.91$). In the previous studies, correlations ranged from 0.74 to 0.89 [19,21–25]. These subscales were not significantly correlated to 'Alcohol/drugs' subscale, as found in the original version [19], the Greek physicians' version [23] and the Australian healthcare students' version [25]. Correlations between the amount of previous IPV training and the "Background" subscales were moderate and higher than those found in all previous studies [19,21–25]. 'Actual knowledge' subscale had small correlations with the amount of

previous training, the "Background" and "Opinions" subscales. The higher correlation was found with the 'Victim understanding' subscale, which was the only significant correlation in the Greek physicians [23] and American healthcare students' versions [21]. Short et al in the original version found significant but small correlations between 'Actual knowledge' and 'Perceived knowledge', and between 'Actual knowledge' and "Opinions" subscales, except 'Preparation' and 'Legal requirements' subscales [19]. In the American pharmacists' version, 'Actual knowledge' was only significantly correlated to 'Legal requirements' subscale [22].

Test-retest reliability, which is an essential property [53], showed substantial to almost perfect ICCs, except for the 'Alcohol/drugs' opinions subscale, with moderate reliability. Only the Greek physicians validation reported ICCs for test-retest reliability over a 3- to 4-week period [23]. In our study, ICCs for the 'Perceived preparation', 'Perceived knowledge', 'Actual knowledge' and 'Self-efficacy' subscales were higher (0.92, 0.88, 0.69, and 0.83 versus 0.84, 0.78, 0.68, and.078 respectively). For the other similar "Opinions" subscales, our ICCs were lower: 0.65 versus 0.83 for 'Preparation' subscale, 0.78 versus 0.93 for 'Workplace issues', 0.74 versus 0.81 for 'Victim understanding', 0.46 versus 0.79 for 'Alcohol/drugs'.

Our study has some limitations. First, participants were mainly women. This could be explained by the fact that midwives are widely represented in perinatal care occupations. In France in 2017, only 2.6% of midwives were men. As women, these participants certainly felt more concerned about this topic and this could have affected their responses. Almost one quarter of the participants reported no previous IPV training. In the other cross-cultural validations, this proportion varied from 45.7% [21] to 95% [23]. Nevertheless, Short et al reported that 13.5% of their first sample and 31.3% of their second sample used to validate the original PREMIS reported no previous training, which was closer to our results [19]. Besides, it was not possible to distinguish between town and clinic caregivers and those who are regular employees of the hospitals or health care institutions. Practice variations could be explained by the work site, in particular by the presence of protocol for dealing with abused women, IPV referral resources or institutional policies. For the reliability test, 24 respondents completed the retest, a number lower than the 50 subjects recommended by Terwee et al. [38], but higher than in the other validation studies:10 subjects for the Spanish version [24], 18 subjects for the Australian version [25] and 20 subjects for the Greek version [23].

Further studies are needed on sample with more men to confirm the psychometric properties of the questionnaire, and in other settings, like in primary health care physicians. The responsiveness to change, that is the ability of the PREMIS-French to detect changes after IPV training, had to be evaluated, as well as convergent validity, as no validated instruments was validated in French to serve for comparison.

## Conclusion

This study provides evidence of the good psychometric properties of the PREMIS-French when delivered to perinatal care providers to assess their readiness to manage IPV. As the original and the other versions of the PREMIS, the PREMIS-French could be used in several ways: to assess knowledge, attitudes, beliefs, behaviors and skills in order to assess needs that could be addressed during IPV education program; as a pre and post-test to measure the changes over time (spontaneous evolution of practices) or after IPV trainings or interventions; to compare perinatal care providers who had received training and those who did not have it. This is a valid and easy instrument to use which will help to understand perinatal care providers' barriers to IPV screening and management and will help to focus on specific lacks of knowledge for developing IPV education programs. This measure of educational outcome would also

allow the evaluation of IPV training courses that will be developed in the near future, in accordance with the recent recommendations of the French National Health Authority.

## Supporting information

**S1 Table. Sociodemographic and professional characteristics of participants.**
(DOCX)

**S2 Table. Corrected item-total correlations for the PREMIS-French "Opinions" subscales.**
(DOCX)

**S1 File. PREMIS-French questionnaire.**
(DOCX)

## Author Contributions

**Conceptualization:** Candy Guiguet-Auclair, Anne Debost-Legrand, Didier Lémery, Françoise Vendittelli.

**Data curation:** Candy Guiguet-Auclair, Anne Debost-Legrand.

**Formal analysis:** Candy Guiguet-Auclair, Anne Debost-Legrand.

**Investigation:** Anne Debost-Legrand.

**Methodology:** Candy Guiguet-Auclair, Anne Debost-Legrand, Didier Lémery, Chloé Barasinski, Françoise Vendittelli.

**Project administration:** Anne Debost-Legrand.

**Validation:** Candy Guiguet-Auclair, Anne Debost-Legrand.

**Writing – original draft:** Candy Guiguet-Auclair, Anne Debost-Legrand.

**Writing – review & editing:** Candy Guiguet-Auclair, Anne Debost-Legrand, Didier Lémery, Chloé Barasinski, Blandine Mulin, Françoise Vendittelli.

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
