## [Decision Letter · Decision Letter 0]

27 Aug 2021

PONE-D-21-19470

Measuring the readiness to screen and manage intimate partner violence: cross-cultural adaptation and psychometric properties of the PREMIS tool for perinatal healthcare professionals

PLOS ONE

Dear Dr. Guiguet-Auclair,

Thank you for submitting your manuscript to PLOS ONE. After careful consideration, we feel that it has merit but does not fully meet PLOS ONE’s publication criteria as it currently stands. Therefore, we invite you to submit a revised version of the manuscript that addresses the points raised during the review process.

**A thorough and rich review was conducted on the manuscript, which will surely guide the authors to carefully revise the work to make it suitable for publication.**

We look forward to receiving your revised manuscript.

Kind regards,

Stefano Federici, Ph.D.

Academic Editor

PLOS ONE

Journal Requirements:

2. We note the tables in your submission have been adapted from the scales in https://www.ncbi.nlm.nih.gov/pmc/articles/PMC1451776/ and https://www.ajpmonline.org/article/S0749-3797(05)00401-0/fulltext#tables.

Before we can proceed, please clarify if you received explicit written permission from the copyright holder of the scales to publish your tables in PLOS ONE under a CC BY 4.0 license.

To seek permission from the American Journal of Preventative Medicine to publish the scales used in your tables under the Creative Commons Attribution License (CCAL), CC BY 4.0, please contact them please contact them with the following text and the PLOS ONE Request for Permission form (http://journals.plos.org/plosone/s/file?id=7c09/content-permission-form.pdf):

“I request permission for the open-access journal PLOS ONE to publish XXX under the Creative Commons Attribution License (CCAL) CC BY 4.0 (http://creativecommons.org/licenses/by/4.0/). Please be aware that this license allows unrestricted use and distribution, even commercially, by third parties. Please reply and provide explicit written permission to publish XXX under a CC BY license.”

Please upload the granted permission to the manuscript as a supporting information file. In the table captions where the scales are used, please include the following text: “Republished from [ref] under a CC BY license, with permission from [name of publisher], original copyright [original copyright year].”

Please note that RightsLink permission forms often impose use restrictions that are incompatible with our CC BY 4.0 license, and we are therefore unable to accept these permissions. For this reason, we strongly recommend contacting copyright holders with the PLOS ONE Request for Permission form.

If you are unable to obtain permission from the journal, please either A) remove the tables or B) link to or refer to the previously published scales in your manuscript.

Additional Editor Comments (if provided):

A thorough and rich review was conducted on the manuscript, which will surely guide the authors to carefully revise the work to make it suitable for publication.

Reviewers' comments:

Reviewer's Responses to Questions

**Comments to the Author**

1. Is the manuscript technically sound, and do the data support the conclusions?

Reviewer #1: Yes

2. Has the statistical analysis been performed appropriately and rigorously? 

Reviewer #1: Yes

3. Have the authors made all data underlying the findings in their manuscript fully available?

Reviewer #1: Yes

4. Is the manuscript presented in an intelligible fashion and written in standard English?

Reviewer #1: Yes

5. Review Comments to the Author

Reviewer #1: Thank you very much for giving me the opportunity to review this original article based on the analysis of 360 responses to a questionnaire translated by the authors.

The authors' objectives were to translate an existing questionnaire and then describe its psychometric properties.

After some revisions, this article deserves to be published in my opinion as it makes an important contribution to the current literature. Indeed, to my knowledge, there is no validated questionnaire in French to measure the readiness/capacity of health care workers to manage intimate partner violence.

Regarding the title, the authors have chosen to use the term "cross-cultural adaptation" whereas I would describe it more as a translation. The authors have chosen to present the method and the result of this translation in the method part of their article. Consequently, it seems to me that the main objective of this article is not to present the translation of the questionnaire but rather to describe its psychometric properties. Please justify the choice of using the term "cross-cultural adaptation" in the title by specifying how you did not only translate but also culturally adapt the original questionnaire or please edit the title.

Regarding the abstract, the introduction part could be reduced to two sentences. There is a gap between the two announced aims (of translating the questionnaire and then describing its psychometric properties) and the first sentence of the results which talks about "acceptability". In the scenario where the authors choose to keep translation as the main aim of this study, I recommend that the authors start the results directly by talking about the translation (and the addition of one item and the deletion of another) and then the psychometric results (data completeness, factor analysis, score distribution, floor and ceiling effects, internal consistency, item-total correlations, inter-subscale correlations and test-retest reliability). The questionnaire consists of 5 parts including 9 (?) subscales but the subscale "Actual IPV Knowledge" is not presented in the abstract. Please explain why or revise it.

Regarding the introduction part:

- page 3, line 53, delete the 'r' between "factors" and "of IPV".

- page 3, line 63, I am confused by the wording "perinatal healthcare professionals" which I would be inclined to translate as "healthcare professional specialising in the perinatal period". Please check with an experienced linguist.

- page 4, line 76, specify whether or not the questionnaire ref 14 has been validated.

Regarding the methods part:

It is not clear to me how the authors were able to identify participants, avoid multiple responses from the same person and, conversely, make the link between test and retest.

- page 6, line 138, replace "IPV Knowledge" by "Actual IPV Knowledge"

- page 9, line 196, replace "date" by "data"

As I am not a specialist in factor analysis according to KMO, I am not able to comment on this part. The other sections of the method are clear and do not call for any further comment.

Regarding the results part:

Do the authors have data on the response rate, or on the rate of completed questionnaires in relation to those started?

The tables are numerous (7) but clear and useful. Perhaps a simplified version of table 1 could be presented in the article and the full table put in an appendix. In any case, it would be interesting to have the characteristics of the 24 re-test participants in this table. Similarly, the authors should consider putting other tables in the appendix, such as Table 4. Please explain why you have separated the inter-subscale correlations analysis into two tables 5 and 6 and included the number of hours of previous IPV training?

- page 14, line 294, add a bracket after "Table 2".

- page 397, you mention the mean age and its standard deviation but with 24 responses, does this follow a normal distribution? If not, the median and the 25th and 75th percentiles would be more appropriate.

Do not put spaces before and after the slash (/).

Regarding the discusion part:

- page 9, line 418, The authors report "good acceptability" due to a "high response rates" although the response rates are unknown. A total of 360 responses were obtained but the deonominator is not reported in this article. To how many healthcare professionals was the questionnaire sent? What are the response rates? These considerations also need to be clarified for the re-test where 24 persons responded but where the denominator is unknown. Do the authors know the proportion of men in these two populations (to be able to compare them to the proportions of their two samples)?

Although a little long, the discussion part is relevant and the limitations of the study are fairly presented.

Regarding the Annex (S1 File.docx):

- page 1, question 2, please consider replacing "Sexe" with "Genre";

- page 1, question 4, I recommend that you add "Maïeutique" after "Gynécologie / Obstétrique";

- page 1, question 5, I recommend that you reword the sentence as follows: "En quelle année avec-vous obtenu votre diplôme d'exercice professionnel ?";

- page 2, question 1, there is an extra space after "résidanat/";

- On page 2, question 3b, I understand that the questionnaire has been tested as it is and that a modification after the fact is questionable however please consider replacing "divulgations de maltraitance" by "révélations de maltraitance/abusbecause in French the word "révéler" means "to inform someone of something that was ignored, unknown, hidden or secret", while "divulguer" means "to bring to the attention of a large audience information that was initially considered to be or should remain confidential";

- page 5, question 8, you have translated "Child Protective Services" by "CRIP" without specifying the meaning of this acronym. I suggest that you either use a generic term or specify the acronym;

- page 6, question 4, delete the full stop at the end of the sentence;

- page 7, question 17, there is an extra space between "signalement" and "de". The full stop at the end of the sentence should be removed;

- page 10, question 4, replace "1 avocat" by "un avocat".

Regarding the accessibility of the data, the authors state that the data are fully available without restriction at Mendeley data repository. A digital object identifier (DOI) to easily access this data would be useful.

All my comments may seem too numerous, but for the most part they are minor details that can easily be corrected by the authors. Once again, I would like to highlight the importance of this work which fills a gap in the training of health professionals regarding intimate partner violence. The number of responses is significant for a 25-minute questionnaire administered to professionals. These data are presented with rigour by the authors and deserve to be published in my opinion.

6. PLOS authors have the option to publish the peer review history of their article (what does this mean?). If published, this will include your full peer review and any attached files.

Reviewer #1: **Yes: **Laurent Gaucher

---

## [Author Response · Author response to Decision Letter 0]

16 Sep 2021

Journal Requirements

Response: We ensure that our manuscript meets PLOS ONE’s style requirements, using the PLOS ONE style templates. 

2. We note the tables in your submission have been adapted from the scales in https://www.ncbi.nlm.nih.gov/pmc/articles/PMC1451776/

and https://www.ajpmonline.org/article/S0749-3797(05)00401-0/fulltext#tables. If you are unable to obtain permission from the journal, please either A) remove the tables or B) link to or refer to the previously published scales in your manuscript.

Response: Tables 5 and 6 in our first manuscript are adapted from Tables 3 and 2 respectively in the validation of the original PREMIS by Short et al. (Short LM, Alpert E, Harris JM, Surprenant ZJ. A tool for measuring physician readiness to manage intimate partner violence. Am J Prev Med. 2006; 30: 173–180. doi:10.1016/j.amepre.2005.10.009). 

We were unable to obtain permission from the American Journal of Preventive Medicine. So, we added “Adapted from Short et al. [19]” in the footnotes of Tables 5 and 6 (Tables 4 and 5 in the revised manuscript).

b) State what role the funders took in the study. If the funders had no role in your study, please state: “The funders had no role in study design, data collection and analysis, decision to publish, or preparation of the manuscript.”c) If any authors received a salary from any of your funders, please state which authors and which funders.d) If you did not receive any funding for this study, please state: “The authors received no specific funding for this work.”

Response: We did not receive any funding for this study. We precise this in the cover letter. 

Response: We have described the changes made to our Data Availability statement in the cover letter. 

Reviewers' comments

Thank you very much for giving me the opportunity to review this original article based on the analysis of 360 responses to a questionnaire translated by the authors. The authors' objectives were to translate an existing questionnaire and then describe its psychometric properties. 

After some revisions, this article deserves to be published in my opinion as it makes an important contribution to the current literature. Indeed, to my knowledge, there is no validated questionnaire in French to measure the readiness/capacity of health care workers to manage intimate partner violence.

Regarding the title, the authors have chosen to use the term "cross-cultural adaptation" whereas I would describe it more as a translation. The authors have chosen to present the method and the result of this translation in the method part of their article. Consequently, it seems to me that the main objective of this article is not to present the translation of the questionnaire but rather to describe its psychometric properties. Please justify the choice of using the term "cross-cultural adaptation" in the title by specifying how you did not only translate but also culturally adapt the original questionnaire or please edit the title.

Response: In order to be more precise, we have changed the presentation of the translation and cross-cultural adaptation of the PREMIS in the method and results sections. In the method section, we only described the methodology used for the translation and cross-cultural adaptation. Then, in the results section, we added a sub-section “Translation and cross-cultural adaptation” where we detailed the cultural adaptations made to the original PREMIS tool. We also detailed here the results of the pre-testing of the French version (obtained after the forward-backward translations) on a sample of 10 health care providers. In the abstract, we added a sentence at the beginning of the results section: “The PREMIS was successfully translated and cross-culturally adapted into French”. 

We chose to use the term “cross-cultural adaptation” as we did not only translate into French, but also culturally adapt the PREMIS, as recommended in international guidelines. It is important that the translation produces a questionnaire comparable in terms of language but also a questionnaire culturally relevant to the French context. For example, ‘Child Protective Services’ was not simply translated into ‘Services de protection des Enfants’ but was cross-culturally adapted into ‘Cellule de Recueil des Informations Préoccupantes (CRIP)’.

Beaton et al. (Beaton DE, Bombardier C, Guillemin F, Ferraz MB. Guidelines for the process of cross-cultural adaptation of self-report measures. Spine. 2000;25: 3186–3191) suggested that “the items must not only be translated well linguistically, but also must be adapted culturally to maintain the content validity of the instrument at a conceptual level across different cultures (…) The term “cross-cultural adaptation” is used to encompass a process that looks at both language (translation) and cultural adaptation issues in the process of preparing a questionnaire for use in another setting.”

Guillemin et al. (Guillemin F, Bombardier C, Beaton D. Cross-cultural adaptation of Health-Related Quality of Life measures: literature review and proposed guidelines. J Clin Epidemiol 1993;46: 1417–32) stated that “cross-cultural adaptation has two components: the translation of HRQL measure and its adaptation, i.e., a combination of the literal translation of individual words and sentences from one language to another and an adaptation with regards to idiom, and to cultural context and lifestyle. “

Regarding the abstract, the introduction part could be reduced to two sentences. 

Response: We reduced the introduction of the abstract as recommended:

“Pregnancy and perinatal periods are significant risk factors of intimate partner violence (IPV), a major public health problem that could begin or intensify during these periods. Perinatal care providers have a major role in the identifications and the management of IPV. This study aimed to cross-culturally adapt into French the Physician Readiness to Manage Intimate Partner Violence Survey (PREMIS) tool, a reliable instrument to assess the knowledge, attitudes and preparedness to address IPV, and to evaluate its psychometric properties”.

There is a gap between the two announced aims (of translating the questionnaire and then describing its psychometric properties) and the first sentence of the results which talks about "acceptability". In the scenario where the authors choose to keep translation as the main aim of this study, I recommend that the authors start the results directly by talking about the translation (and the addition of one item and the deletion of another) and then the psychometric results (data completeness, factor analysis, score distribution, floor and ceiling effects, internal consistency, item-total correlations, inter-subscale correlations and test-retest reliability). 

Response: In the abstract, as recommended, we added a sentence at the beginning of the results section: “The PREMIS was successfully translated and cross-culturally adapted into French”. 

The questionnaire consists of 5 parts including 9 (?) subscales but the subscale "Actual IPV Knowledge" is not presented in the abstract. Please explain why or revise it.

Response: Short et al. explained in the validation study of the PREMIS that for the ‘Actual Knowledge’ subscale “measurement of internal consistency for this criterion-referenced section of the instrument was not appropriate” and cited Brown et al (Brown JD. The Cronbach alpha reliability estimate. Shiken: JALT Testing & Evaluation SIG Newsletter, February 2002, vol. 6, no. 1, pp.16 –18. Available at: www.jalt.org/test/bro_13.htm).

Brown et al. explained that “Cronbach alpha is appropriately applied to norm-referenced tests and norm-referenced decisions (e.g., admissions and placement decisions), but not to criterion-referenced tests and criterion-referenced decisions (e.g., diagnostic and achievement decisions)”.

We added a sentence in the methods section (page 11, lines 223-224 of the revised manuscript): “Internal consistency was not evaluated for the ‘Actual knowledge’ subscale as it is a criterion-referenced subscale [19,49]”.

Regarding the introduction part:

- page 3, line 53, delete the 'r' between "factors" and "of IPV".

Response: This was done.

- page 3, line 63, I am confused by the wording "perinatal healthcare professionals" which I would be inclined to translate as "healthcare professional specialising in the perinatal period". Please check with an experienced linguist.

Response: We replaced “perinatal healthcare professionals” by “perinatal care providers” in all the manuscript. In the second paragraph of the introduction (page 4, lines 64-65 of the revised manuscript), we first defined perinatal care providers by healthcare professionals specializing in the perinatal period. 

- page 4, line 76, specify whether or not the questionnaire ref 14 has been validated.

Response: We specified that the questionnaire ref 14 has not been validated (page 5, lines 78-79 of the revised manuscript):

“In 2014, the French inter-ministerial mission for the protection of women victims of violence and the fight against human trafficking conducted a national study to evaluate the midwives’ IPV knowledge and their ability to screen IPV with a non-validated questionnaire [14]”.

Regarding the methods part:

It is not clear to me how the authors were able to identify participants, avoid multiple responses from the same person and, conversely, make the link between test and retest.

Response: We were not able to identify precisely participants. The French Federation of Perinatal Heath Network (‘Fédération Française des Réseaux de Santé en Périnatalité’, FFRSP) contacted its care providers’ memberships to participate. We revised this part describing how the participants received the link to complete the questionnaire in the ‘Study design and participants’ sub-section (pages 6-7, lines 121-125 of the revised manuscript): 

“The French Federation of Perinatal Heath Network (‘Fédération Française des Réseaux de Santé en Périnatalité’) invited by email its care providers’ memberships (except healthcare students) to take part in the study and spread the internet link to complete the questionnaire.”

We checked for multiple responses from the same person by crossing variables: age, gender, department of France, field of practice, number of years practicing and number of patients seen per week. 

We revised the part describing how participants to test-retest were recruited (page 7, lines 126-130 of the revised manuscript):

“To assess test-retest reliability of the PREMIS, the internet link to complete the questionnaire a first time was sent by email to a subsample of memberships of two regional perinatal networks who were identified by their email address. They completed a second time the same questionnaire online between fifteen days and one month after the first assessment.”

- page 6, line 138, replace "IPV Knowledge" by "Actual IPV Knowledge"

Response: This was replaced. 

- page 9, line 196, replace "date" by "data"

Response: This was replaced. 

As I am not a specialist in factor analysis according to KMO, I am not able to comment on this part. The other sections of the method are clear and do not call for any further comment.

Regarding the results part:

Do the authors have data on the response rate, or on the rate of completed questionnaires in relation to those started?

Response: Unfortunately we did know the response rate, as the number of perinatal care providers invited to participate by the French Federation of Perinatal Health Network was not known. 

However, this is not a major problem in our study, as our purpose was not to assess the knowledge, attitudes and preparedness to address intimate partner violence, with a representative sample, but was to study the psychometric properties of the French version of the PREMIS. 

A sample size of 100 subjects at least is recommended for a psychometric evaluation by COSMIN (COSMIN. COSMIN - Improving the selection of outcome measurement instruments. 2020. Available: https://www.cosmin.nl/) and Terwee et al. (Terwee CB, Bot SDM, de Boer MR, van der Windt DAWM, Knol DL, Dekker J, et al. Quality criteria were proposed for measurement properties of health status questionnaires. J Clin Epidemiol. 2007;60: 34–42. doi:10.1016/j.jclinepi.2006.03.012). 

A subject-item ratio from 4 to 10 subjects is also often recommended for factor analysis. In our study, 360 subjects provided a subject-item ratio of 10:1 regarding the factor analysis of the 36 items of the “Opinions” section. 

The tables are numerous (7) but clear and useful. Perhaps a simplified version of table 1 could be presented in the article and the full table put in an appendix. In any case, it would be interesting to have the characteristics of the 24 re-test participants in this table. 

Response: As proposed, a simplified version of Table 1 was presented in the article and the full table 1 was put in a supplementary file (S1 Table).The characteristics of the 24 test-retest participants was added in the S1 Table.

Similarly, the authors should consider putting other tables in the appendix, such as Table 4. 

Response: Table 4 was presented in a supplementary file as suggested (S2 Table). 

Please explain why you have separated the inter-subscale correlations analysis into two tables 5 and 6 and included the number of hours of previous IPV training?

Response: We thanks the reviewer for this important remark. We have not been sufficiently clear in our methodology. The inter-subscale correlations analysis was for the “Opinions” section subscales. That is why the results are reported in one table (Table 4 in the revised manuscript). 

The correlations presented in Table 6 (Table 5 in the revised manuscript) assessed the convergent validity of the PREMIS. Correlations between the different subscales themselves and with the amount of previous training were hypothetically attempted. Good convergent validity is demonstrated when expected correlations are found. 

We have modified in the methods section the sub-section “Inter-subscale correlations” and have added a sub-section for the convergent validity analysis. In the results part, we added a sub-section title for the convergent validity. 

- page 14, line 294, add a bracket after "Table 2".

Response: This was done.

- page 397, you mention the mean age and its standard deviation but with 24 responses, does this follow a normal distribution? If not, the median and the 25th and 75th percentiles would be more appropriate.

Response: We checked that the distribution of age among the 24 responses follow a normal distribution (Shapiro-Wilk test: p=0.3408). For this reason, mean age and its standard deviation was reported.

Do not put spaces before and after the slash (/).

Response: This was done.

Regarding the discusion part:

- page 9, line 418, The authors report "good acceptability" due to a "high response rates" although the response rates are unknown. 

Response: In the second paragraph of discussion part, we replaced “high response rates” by “low percentages of missing values per item”, as response rate could lead to confusion (page 29, line 437 of the revised manuscript). 

A total of 360 responses were obtained but the deonominator is not reported in this article. To how many healthcare professionals was the questionnaire sent? What are the response rates?

Response: Unfortunately we did know the response rate. As mentioned above, the number of perinatal care providers invited to participate by email by the French Federation of Perinatal Health Network was not known. However, as previously specified, recommendations for psychometric evaluation of 100 subjects at least and of subject-item ratio from 4 to 10 were achieved. 

These considerations also need to be clarified for the re-test where 24 persons responded but where the denominator is unknown. 

Response: A sentence concerning the response rate for test-retest was added in the Reliability subsection (page 27, lines 413-414 of the revised manuscript).

“Of the fifty perinatal care providers randomly selected for the test-retest, 40 (80.0%) completed the test questionnaire. Among them, 24 (60.0%) completed the retest questionnaire”. 

Do the authors know the proportion of men in these two populations (to be able to compare them to the proportions of their two samples)?

Response: We did not know the proportion of men among the memberships of the French Federation of Perinatal Health Network and among the memberships of the two regional perinatal networks (sample for test-retest reliability). Unfortunately, the FFRSP (‘Fédération Française des Réseaux de Santé en Périnatalité’) did not produce statistics on sociodemographics and clinical characteristics of their memberships. 

We only found that in 2017, the proportion of men among midwives was 2.6% (https://www.ordre-sages-femmes.fr/etre-sage-femme/donnees-demographiques-de-la-profession/). We added this information in the discussion (page 32, lines 517-518 of the revised manuscript):

“In France in 2017, only 2.6% of midwives were men”.

Although a little long, the discussion part is relevant and the limitations of the study are fairly presented.

Regarding the Annex (S1 File.docx):

- page 1, question 2, please consider replacing "Sexe" with "Genre";

Response: This was replaced.

- page 1, question 4, I recommend that you add "Maïeutique" after "Gynécologie / Obstétrique";

Response: “Maïeutique” was added. 

- page 1, question 5, I recommend that you reword the sentence as follows: "En quelle année avec-vous obtenu votre diplôme d'exercice professionnel ?";

Response: The sentence was reworded as recommended.

- page 2, question 1, there is an extra space after "résidanat/";

Response: This was corrected.

- On page 2, question 3b, I understand that the questionnaire has been tested as it is and that a modification after the fact is questionable however please consider replacing "divulgations de maltraitance" by "révélations de maltraitance/abusbecause in French the word "révéler" means "to inform someone of something that was ignored, unknown, hidden or secret", while "divulguer" means "to bring to the attention of a large audience information that was initially considered to be or should remain confidential";

Response: The term “disclosures” was first translated by “révélations” in the forward translations. The multidisciplinary committee decided to modify “révélations” by “divulgations”. This term “divulgations” was correctly backward translated in “disclosures”. During the pre-test study, no difficulties concerning this item was reported and no comments as well. 

- page 5, question 8, you have translated "Child Protective Services" by "CRIP" without specifying the meaning of this acronym. I suggest that you either use a generic term or specify the acronym;

Response: We have specified the acronym: 

« Si l'enfant ne court pas un danger immédiat, les professionnels de santé ne doivent pas effectuer un signalement de cas d’enfants témoin de violence conjugale auprès de la Cellule de Recueil des Informations Préoccupantes (CRIP) »

- page 6, question 4, delete the full stop at the end of the sentence;

Response: This was deleted.

- page 7, question 17, there is an extra space between "signalement" and "de". The full stop at the end of the sentence should be removed;

Response: This was corrected.

- page 10, question 4, replace "1 avocat" by "un avocat".

Response: This was replaced.

Regarding the accessibility of the data, the authors state that the data are fully available without restriction at Mendeley data repository. A digital object identifier (DOI) to easily access this data would be useful.

Response: We made changes to our Data Availability statement: 

All data are available at Mendely: Guiguet-Auclair, Candy; Debost-Legrand, Anne; Lémery, Didier; Barasinski, Chloé; Mulin, Blandine; Vendittelli, Françoise (2021), “Measuring the readiness to screen and manage intimate partner violence : cross-cultural adaptation and psychometric evaluation of the PREMIS tool for perinatal care providers”, Mendeley Data, V1, doi: 10.17632/7x989dtpk7.1.

---

## [Decision Letter · Decision Letter 1]

5 Oct 2021

PONE-D-21-19470R1Measuring the readiness to screen and manage intimate partner violence: cross-cultural adaptation and psychometric evaluation of the PREMIS tool for perinatal care providersPLOS ONE

Dear Dr. Guiguet-Auclair,

Thank you for submitting your manuscript to PLOS ONE. After careful consideration, we feel that it has merit but does not fully meet PLOS ONE’s publication criteria as it currently stands. Therefore, we invite you to submit a revised version of the manuscript that addresses the points raised during the review process.

**Just one more minor revision and then the manuscript will go to the Acdemic Editor's decision without further review by the Reviewer.**

We look forward to receiving your revised manuscript.

Kind regards,

Stefano Federici, Ph.D.

Academic Editor

PLOS ONE

Journal Requirements:

Additional Editor Comments (if provided):

Just one more minor revision and then the manuscript will go to the Acdemic Editor's decision without further review by the Reviewer.

Reviewers' comments:

Reviewer's Responses to Questions

**Comments to the Author**

1. If the authors have adequately addressed your comments raised in a previous round of review and you feel that this manuscript is now acceptable for publication, you may indicate that here to bypass the “Comments to the Author” section, enter your conflict of interest statement in the “Confidential to Editor” section, and submit your "Accept" recommendation.

Reviewer #1: All comments have been addressed

2. Is the manuscript technically sound, and do the data support the conclusions?

Reviewer #1: Yes

3. Has the statistical analysis been performed appropriately and rigorously? 

Reviewer #1: Yes

4. Have the authors made all data underlying the findings in their manuscript fully available?

Reviewer #1: Yes

5. Is the manuscript presented in an intelligible fashion and written in standard English?

Reviewer #1: Yes

6. Review Comments to the Author

Reviewer #1: Many thanks to the authors for taking my comments into account.

Regarding the abstract, I would substitute the following sentence "Results: The PREMIS was successfully translated and cross-culturally adapted into French." by "Results: The PREMIS was successfully translated and cross-culturally adapted to the context of metropolitan France.". Indeed, French is spoken in many countries but in different contexts.

I have no further comments and again I thank the authors for their conscientious work.

7. PLOS authors have the option to publish the peer review history of their article (what does this mean?). If published, this will include your full peer review and any attached files.

Reviewer #1: **Yes: **Laurent GAUCHER

---

## [Author Response · Author response to Decision Letter 1]

8 Oct 2021

Journal Requirements

Response: We have reviewed all the references in the manuscript and we have modified the formatting of some of them, in order to be conform with the “Vancouver” style. 

Reviewer' comments

Regarding the abstract, I would substitute the following sentence "Results: The PREMIS was successfully translated and cross-culturally adapted into French." by "Results: The PREMIS was successfully translated and cross-culturally adapted to the context of metropolitan France.". Indeed, French is spoken in many countries but in different contexts.

Response: We modified the first sentence of the Results section in the abstract as recommended.

---

## [Editor Report · Decision Letter 2]

11 Oct 2021

Measuring the readiness to screen and manage intimate partner violence: cross-cultural adaptation and psychometric evaluation of the PREMIS tool for perinatal care providers

PONE-D-21-19470R2

Dear Dr. Guiguet-Auclair,

We’re pleased to inform you that your manuscript has been judged scientifically suitable for publication and will be formally accepted for publication once it meets all outstanding technical requirements.

Kind regards,

Stefano Federici, Ph.D.

Academic Editor

PLOS ONE
---

## [Editor Report · Acceptance letter]

27 Oct 2021

PONE-D-21-19470R2 

Measuring the readiness to screen and manage intimate partner violence: cross-cultural adaptation and psychometric evaluation of the PREMIS tool for perinatal care providers 

Dear Dr. Guiguet-Auclair:

I'm pleased to inform you that your manuscript has been deemed suitable for publication in PLOS ONE. Congratulations! Your manuscript is now with our production department. 

Kind regards, 

on behalf of

Prof. Stefano Federici 

Academic Editor

PLOS ONE